# Phylogeography and Population Demography of *Parrotia subaequalis*, a Hamamelidaceous Tertiary Relict ‘Living Fossil’ Tree Endemic to East Asia Refugia: Implications from Molecular Data and Ecological Niche Modeling

**DOI:** 10.3390/plants14121754

**Published:** 2025-06-07

**Authors:** Yunyan Zhang, Zhiyuan Li, Qixun Chen, Yahong Wang, Shuang Wang, Guozheng Wang, Pan Li, Hong Liu, Pengfu Li, Chi Xu, Zhongsheng Wang

**Affiliations:** 1College of Life Sciences, Nanjing University, Nanjing 210023, China; zyynju@nju.edu.cn (Y.Z.); 502023300067@smail.nju.edu.cn (Z.L.); 502024300060@smail.nju.edu.cn (Q.C.); 502024300072@smail.nju.edu.cn (Y.W.); mg20300068@smail.nju.edu.cn (S.W.); mt213670@163.com (G.W.); pengfuli@nju.edu.cn (P.L.); 2College of Life Sciences, Sichuan University, Chengdu 610065, China; 3Systematic & Evolutionary Botany and Biodiversity Group, MOE Laboratory of Biosystem Homeostasis and Protection, College of Life Sciences, Zhejiang University, Hangzhou 310058, China; panli_zju@126.com; 4Department of Earth and Environment, Florida International University, Westchester, FL 33199, USA; hliu@fiu.edu

**Keywords:** molecular phylogeography, population demography, genetic diversity and differentiation, glacial refugia, *Parrotia subaequalis*, Hamamelidaceae, Tertiary relict ‘living fossil’ tree

## Abstract

The diverse topography and mild monsoon climate in East Asia are considered to be important drivers for the long-term ecological success of the Tertiary relict ‘living fossil’ plants during the glacial/interglacial cycles. Here we investigated the phylogeographic pattern and demographic history of a hamamelidaceous Tertiary relict ‘living fossil’ tree (*Parrotia subaequalis*) endemic to the subtropical forests of eastern China, employing molecular data and ecological niche modeling. In the long evolutionary history, *P. subaequalis* has accumulated a high haplotype diversity. Weak gene flow by seeds, geographical isolation, and heterogeneous habitats have led to a relatively high level of genetic differentiation in this species. The divergence time of two cpDNA lineages of *P. subaequalis* was dated to the late Miocene of the Tertiary period, and the diversification of haplotypes occurred in the Quaternary period. Paleo-distribution modeling suggested that *P. subaequalis* followed the pattern of ‘glacial expansion-interglacial compression’. The Dabie Mountain and Yellow Mountain in Anhui Province and the Tianmu Mountain and Simin Mountain in Zhejiang Province were inferred to be multiple glacial refugia of *P. subaequalis* in East Asia and have been proposed to be protected as ‘Management Units’. Collectively, our study offers insights into the plant evolution and adaptation of *P. subaequalis* and other Tertiary relict ‘living fossil’ trees endemic to East Asia refugia.

## 1. Introduction

Currently, within some warm and humid refugial zones in the temperate biome of Eurasia (i.e., East Asia, Greece, Italy, and Iran), some Tertiary relict ‘living fossil’ plant taxa reside through the geological transition and climatic oscillation [1,2,3]. Among these glacial refugia, the East Asian flora (EAF) harbors the highest number of species and is a key biodiversity hotspot because it is not heavily covered by extensive glaciers [2,4]. The subtle influence of glaciers, the upliftment of the Himalayan–Tibetan Plateau, the aridification of Central Asia, and the formation of Asian monsoons are all considered to have shaped today’s diverse phylogeographic patterns of taxa in EAF [5,6,7]. Recently, researchers have divided the EAF into two main floras: the *Rhododendron* flora and the *Metasequoia* flora. The *Metasequoia* flora is rich in ancient plant lineages in subtropical China (stretching from the eastern Tibetan Plateau to the Pacific Ocean and from the Qingling Mountains-Huai River line (at ca. 34° N) to the tropical south (≤22° N)) [4,6]. The diverse topography and mild monsoon climate in subtropical China are thought to be important drivers for the long-term sustained ecological success of the rare Tertiary relict plant lineages during the glacial/interglacial cycles [3,8].

Previous phylogeographic research on the *Metasequoia* flora in subtropical China suggested the existence of multiple tree refugia (MR hypothesis), which provided excellent local micro-environments and climatic ecological niches for various relict species and were proposed as hotspots for biodiversity conservation [7,9,10,11]. Driven by climatic changes and geo-historical isolations, these Eurasian relics experienced the vicariance events (e.g., continental disjunctions, allopatric speciation) and repeated demographic and range expansions and contractions [12,13]. However, they were capable of tracking suitable ecological niches or accumulating genetic variations to adapt to their changing habitats and thus undergo adaptive evolution in the context of global climate change [14,15].

Here, our focal “Tertiary relict tree”, *Parrotia subaequalis* (*Parrotia*, Hamamelidaceae), is a monoecious, wind-pollinated and dispersed deciduous tree; it is characterized by a flecked trunk, exfoliating bark, obovate leaves with colorful autumn tints, apetalous bisexual flowers with linear filaments, and two-celled sub-globose horned capsules [16,17]. Currently, *P. subaequalis* is endemic to the fragmented montane sites of the subtropical forests in the Anhui, Jiangsu, Zhejiang, and Henan provinces of eastern China, with small populations growing on mountain slopes or in ravines [16,17]. However, the fossil records of *Parrotia* indicate that this genus was widely distributed in Asia and Europe during the epoch of the Miocene period (i.e., 23.03–5.333 Mya) [18,19]. Such relict plant lineages are of considerable importance in the evolution and conservation of biodiversity in their changing habitats [20]. It has been listed as a “National Grade I Key Protected Wild Plant” and “Plant Species with Extremely Small Populations, PSESP” in China [21,22]. Apart from the elegant form of this plant used for the horticultural ornamental tree, *P. subaequalis* also has important medicinal and ecological values: Aerial parts (leaf and stem) of *P. subaequalis* contain tannin, salidroside, aglucone, and other medicinal chemical components; this relict tree has strong abilities to resist environmental stresses such as arid land, saline and alkaline land, and limestone mountains [16,17,23]. So far, there have been some small-scale investigations of its population diversity by employing the molecular markers (AFLP, EST-SSR, nSSR) for *P. subaequalis* [17,24,25]. However, little is known about the phylogeographic pattern and population structure and genetic diversity of this living fossil across the entire species distribution range by using informative DNA markers.

The maternally inherited chloroplast DNA (cpDNA) in *P. subaequalis* provides information on past changes in species distribution that are unaffected by subsequent pollen flow, whereas nuclear genetic diversity and structure would be shaped by the colonization of new habitats via both pollen flow and seed dispersal [26]. Therefore, the joint use of organelle and nuclear markers can provide a more comprehensive description of the population structure and provide insights into population history and dynamics [27,28]. In this study, we employed both plastid (cpDNA) and nuclear data (EST-SSRs) to illuminate the range-wide genetic diversity, structure and phylogeographic pattern of *P. subaequalis*. Furthermore, by performing ecological niche modelling (ENM) across temporal scales, we can predict the species’ potential distribution during the paleo-period, present, and future [29] and investigate the responses to climatic oscillations, which have implications for the conservation of *P. subaequalis*.

Collectively, *P. subaequalis* inhabiting the fragmented montane sites of subtropical forests represents a promising system for disentangling whether the long-term ecological success of this Tertiary relict tree could be explained by the MR hypothesis and thus shed light on the evolution, adaptation, and conservation of *P. subaequalis* and other Tertiary relict trees.

## 2. Materials and Methods

### 2.1. Sample Collection, DNA Extraction, cpDNA Sequencing, and EST-SSR Genotyping

Leaf samples of a total of 410 individuals from 21 natural populations of *P. subaequalis* throughout its distribution range in subtropical forests of eastern China were collected (Figure 1; Appendix A). We surveyed a total of 339 individuals from all 21 populations of *P. subaequalis* for cpDNA sequence variations, and 410 individuals from 21 natural populations were employed for the genotyping of EST-SSR loci (Appendix A). *Parrotia persica* (DC.) C. A. Meyer was used as outgroup for the genealogical cpDNA analysis because of its sister relationship in phylogeny [30].

Total genomic DNA was extracted from silica-gel-dried leaf tissues using a modified CTAB-based method [31]. Three intergenic spacer (IGS) primers (*psbC*-*psbZ*, *accD*-*psaI*, and *ndhD*-*psaC*) were selected from the ten divergence hotspot regions of the cpDNA of *P. subaequalis* reported in Zhang et al. (2018) [17] with high polymorphism and capable of stably amplifying with a single clear band. The detailed sequences of these primers are listed in Appendix A. Polymerase chain reaction (PCR) amplifications were performed on a GeneAmp9700 DNA Thermal Cycler (Perkin-Elmer, Waltham, MA, USA) following the standard protocol of the AmpliTaq Gold 360 Master PCR kit (Applied Biosystems, Foster City, CA, USA); total reaction volume was 50 µL containing 1 µL (50 ng) of template DNA, 15 µL of AmpliTaq Gold^TM^ 360 Master Mix (Applied Biosystems, Foster City, CA, USA), 20 µL of double distilled H_2_O, 2 µL of reverse primers (10 µM), and 2 µL of forward primers (10 µM). PCR running procedure was as follows: 5 min initial denaturation at 95 °C, 35 cycles of 45 s at 95 °C, 45 s annealing at an optimal primer temperature (55 °C for *accD*-*psaI*, 56 °C for *psbC*-*psbZ* and *ndhD*-*psaC*), and 90 s synthesis at 72 °C, ending with a 10 min extension at 72 °C and a 4 °C maintained temperature. The PCR products were checked with agarose gels stained by GelRed (Tsingke Biological Technology, Hangzhou, China). Sequences were generated with an ABI 3730XL DNA Analyzer (Applied Biosystems, Foster City, CA, USA) and then assembled, edited, and aligned in Geneious v11.1.2 (https://www.geneious.com/; accessed on 10 October 2024). An inversion of 6 bp found in the *accD*-*psaI* region of three samples was coded as a single binary character, comparable to an indel [32]. All variable sites were included and weighted equally. All new cpDNA sequences in this study were deposited into GenBank with accession numbers PV685069–PV685081.

Sixteen EST-SSR loci with higher polymorphism were selected from the developed markers for *P. subaequalis* from its Illumina-based transcriptome datasets in Zhang et al. (2019) [25] and employed for genotyping (Appendix A). The PCR was conducted using the same thermal cycler in a total volume of 15 μL containing 1 µL (50 ng) of template DNA, 7.5 µL of AmpliTaq Gold^TM^ 360 Master Mix (Applied Biosystems, Foster City, CA, USA), 5.5 µL of deionized water, 0.5 µL of forward primers (10 µM), with the 5′-end labeled with a fluorescent dye (FAM, HEX, TAMRA, or ROX), and 0.5 µL of reverse primers (10 µM). PCR running procedure was as follows: 5 min initial denaturation at 95 °C, 35 cycles of 45 s at 95 °C, 30 s annealing at an optimal primer temperature (Appendix A), and 30 s synthesis at 72 °C, followed by a final 10 min extension step at 72 °C and a 4 °C maintained temperature. The PCR products were detected by an ABI 3730XL capillary electrophoresis analyzer (Applied Biosystems, Foster City, CA, USA) with GeneScan-500LIZ as an internal reference (Applied Biosystems). EST-SSR alleles and genotypes were scored using GeneMarker v2.2.0 (SoftGenetics, State College, PA, USA) and manually checked to reduce genotyping errors.

### 2.2. Population Genetic, Phylogeographic, and Demographic Analyses of Chloroplast DNA Markers

For the chloroplast DNA (cpDNA) data set, the number of haplotypes, haplotype diversity (*h*), and nucleotide diversity (π) for each population of *P. subaequalis* and the species overall were calculated using DnaSP v6.12.01 [33]. The relationships between the genetic diversity (*h* and π) and geographic parameters (latitude and longitude) were evaluated in R v4.0.5 (R Development Core Team, Vienna, Austria, 2021) by conducting linear regression and Pearson correlation analysis. *G*_ST_ (coefficient of genetic variation over all populations of *P. subaequalis*) and *N*_ST_ (coefficient of genetic variation influenced by both haplotype frequencies and genetic distances between haplotypes of *P. subaequalis*) were calculated by the program Permut v1.0 [34]. The amount of variation among the populations of *P. subaequalis* within a region and within a population was calculated by the hierarchical analysis of molecular variance (AMOVA) framework carried out using Arlequin v3.5.2 [35] with significance of *Φ*-statistics tested by 1000 non-parametric random permutations. Population differentiation was also quantified with non-hierarchical analysis of molecular variance by estimating *F*_ST_ among populations of *P. subaequalis*. To examine the effect of geographic distance on genetic structure and the relative contribution of gene flow and drift to genetic structure [36], isolation-by-distance (IBD) analysis for *P. subaequalis* was performed by employing the software GenAlEx v6.502 [37].

Then, the haplotype network was constructed using the Median Joining model in Network v10.2.0.0 [38] to detect intraspecific relationships of cpDNA haplotypes of *P. subaequalis*. Phylogenetic cpDNA haplotype trees of *P. subaequalis* were constructed using maximum likelihood (ML) methods in RAxML v8.2.12 [39] under the best-fitting substitution model (HKY + I) selected by jModelTest v2.1.6 based on the corrected Akaike information criterion (AIC) [40] and Bayesian inference (BI) methods in MrBayes v3.2.6 [41] with same substitution model. Additionally, Bayesian analysis was conducted using two separate runs of the Markov chain Monte Carlo (MCMC) algorithm for 1 million generations and tree sampling every 1000 generations. The first 25% of sampled trees were discarded as burn-in, and the 25% best-scoring trees were used to construct the consensus tree and to estimate the posterior probabilities (PPs). Convergence was determined by estimating the average standard deviation of the split frequencies (<0.01). *P. persica* was selected as outgroup for both trees. Divergence dating analysis was conducted on the cpDNA data set under a Bayesian approach in BEAST v2.6.7 [42] using the HKY + I substitution model and the uncorrelated lognormal relaxed clock. A secondary calibration point inferred from the result of a fossil-based phylogeny of Hamamelidaceae [43] was used to constrain the root node of the phylogenetic tree (Mean, 20.7 Ma; 95% highest posterior density [HPD], 9.7–31.2 Ma). We then used a coalescent tree prior with Bayesian skyline model, and MCMC runs of 10 million generations were performed with sampling every 1000 generations, following a burn-in of the initial 10% cycles. Besides, MCMC samples were inspected in Tracer v1.7.1 [44] to confirm sampling adequacy and convergence of the chains to a stationary distribution. Topologies of the above phylogenetic trees were visualized and edited in FigTree v1.4.4 software.

The demographic history of *P. subaequalis* populations was investigated via neutrality tests and mismatch distribution analysis (MDA) in Arlequin v3.5.2 [35]. Neutrality tests (Tajima’s *D* and Fu’s *FS)* were performed to test potential population expansion/contraction according to the significant negative/positive values of *D* and *FS*. MDA also can infer the past demographic expansion/contraction of species; multimodal mismatch distributions of differences between pairs of haplotypes are expected for populations at demographic equilibrium/contraction, whereas unimodal distributions are expected for populations that have experienced recent demographic expansions [45,46].

### 2.3. Population Genetic Diversity, Structure, and Demographic Analyses of EST-SSRs

For the EST-SSR data sets, we calculated the number of alleles (*N*_A_), observed heterozygosity (*H*_O_), expected heterozygosity (*H*_E_), and polymorphism information content (PIC) for each locus using Cervus v3.0.7 [47]. The inbreeding coefficient (*F*_IS_), genetic differentiation coefficient (*F*_ST_), total genetic diversity (*H*_T_) across the populations for each locus, and average genetic diversity within populations (*H*_S_) were calculated using FSTAT v2.9.4 [48]. Allelic richness (*R*_S_) and private allelic richness were corrected for sample size differences using HP-rare v1.1 [49] with rarefied subsamples of six individuals. Deviations from Hardy–Weinberg equilibrium (HWE) and linkage disequilibrium (LD) for each locus were tested in Genepop v4.7 [50] using the Bonferroni method [51], and all loci were also checked for frequencies of null alleles using FreeNA [52]. For each pair of populations of *P. subaequalis*, the following genetic diversity parameters were estimated across all EST-SSR loci in FSTAT v2.9.4 [48] and GenAlEx v6.5 (Canberra, Australia) [37]: the number of alleles (*N*_A_), observed heterozygosity (*H*_O_), expected heterozygosity (*H*_E_), total genetic diversity (*H*_T_), allelic richness (*R*_S_), and inbreeding coefficient (*F*_IS_). The relationships between the genetic diversity estimates (*N*_A_, *H*_E_, and *R*_S_) and different geographic parameters (latitude and longitude) were evaluated in R v4.1.3 (R Core Team, 2020; https://www.r-project.org/; accessed on 5 December 2024) by calculating Pearson’s product–moment correlations.

To illuminate the genetic structure of *P. subaequalis* in the entire EST-SSR data set, we first conducted the Bayesian clustering analysis using Structure v2.3.4 [53]. Ten runs of Structure were performed with the number of clusters (*K*) varying from 1 to 21, a burn-in length of 10,000, and a run length of 100,000 MCMC replications for each value of *K*. The output files were then uploaded to the online website of Structure Harvester (https://github.com/dentearl/structureHarvester/; accessed on 10 December 2024) [54] to estimate the optimal value of *K* by calculating the posterior probability of the data for a given *K* (Ln *P*(*D*)) [53] and the value of a parameter (Δ*K*) that is the rate of change in the log probability of the data between successive *K* values [55]. Clumpp v1.1.2 [56] was used for data integration, and the software Distruct v1.1 [57] was used to visualize the integration results. Genetic distances among populations were then calculated from allele frequencies and subjected to principal coordinate analysis (PCoA) using GenAlEx v6.5 [37].

Additionally, analysis of molecular variance (AMOVA; Excoffier, Smouse, and Quattro 1992) was performed across all loci in Arlequin v3.5.2 [35] with 1000 permutations to assess the significance of the covariance components. Patterns of isolation-by-distance (IBD) [58] were evaluated at the range-wide scale of *P. subaequalis* using Mantel tests in GenAlEx v6.5 [37]: the values of pairwise *F*_ST_/(1 − *F*_ST_) were regressed against the logarithm of pairwise geographic distance of populations. Bottleneck effect at the population level of *P. subaequalis* was detected using Wilcoxon sign-rank test under three models (infinite allele model (IAM), stepwise mutation model (SMM), and two-phased model (TPM)) in Bottleneck v1.2.02 [59]. A mode shift model was also employed to test for the bottleneck effect in each population of *P. subaequalis* in Bottleneck v1.2.02 [59].

### 2.4. Ecological Niche Modeling

Assuming *P. subaequalis* has not changed (and will not change) its climatic preference over at least the last glacial/interglacial cycle and in the future, potential distribution predictions for *P. subaequalis* at present (1950–2000), Mid-Holocene (MH, *c*. 6 kya), Last Glacial Maximum (LGM, *c*. 21 Kya), Last Interglacial (LIG, *c*. 130 Kya), and future (2080) were predicted using Maxent v3.4.4 with ecological niche modeling (ENM) [60,61]. Information on the geographic distribution of *P. subaequalis* was based on 31 nonredundant presence records, including 21 sampling sites in this study (Appendix A) and 10 records from Global Biodiversity Information Facility (GBIF, http://www.gbif.org/; accessed on 30 December 2024) and Chinese Virtual Herbarium (CVH, https://www.cvh.ac.cn/; accessed on 30 December 2024). Climate layers for 19 bioclimatic variables (Appendix A) of the above periods were downloaded from the WorldClim v1.4 data set (http://www.worldclim.org; accessed on 30 December 2024) [62] at a spatial resolution of 2.5 arcmin.

To avoid ENM overfitting and reduce the influence of correlations among climate variables, we selected the following 11 bioclimatic variables by calculating the correlation between all pairs of 19 bioclimatic parameters from the geographic localities of the species occurrence using SPSS v21 (Pearson’s *r* < 0.9): isothermality (BIO3), temperature seasonality (BIO4), max temperature of warmest month (BIO5), mean temperature of wettest quarter (BIO8), mean temperature of driest quarter (BIO9), mean temperature of coldest quarter (BIO11), annual precipitation (BIO12), precipitation of wettest month (BIO13), precipitation seasonality (BIO15), precipitation of driest quarter (BIO17), and precipitation of warmest quarter (BIO18). Moreover, considering *P. subaequalis* is a very small population species with a narrow range of distribution in East China, for the accuracy of the ENM predictions, the above world climate layers were cropped into the range of 108–125° E and 20–38° N for subsequent analyses via ArcGIS 10.2 (ESRI, Inc., Redlands, CA, USA; https://www.esri.com).

The mean value of 10 replicate models was applied as a result of the distributions. The model was projected into the paleoclimate data simulated by the Community Climate System Model v4.0 (MH, LGM, and LIG) and four future greenhouse gas emission scenarios under the representative concentration pathways (RCP2.6, RCP4.5, RCP6.0, and RCP8.5) to infer the potentially suitable habitat of *P. subaequalis* during the above periods by implementing the options of “Create response curves” and “Do jackknife to measure variable importance of bioclimatic factors”. The accuracy of ENM prediction was assessed by calculating the AUC value (area under the receiver operating characteristic curve) in Maxent [63], where scores between 0.7 and 0.9 suggest good performance of the model, and higher values represent a rather high accuracy [64]. Moreover, seafloor topography data (ETOPO1) from the National Geophysical Data Center of National Oceanic and Atmospheric Administration (NOAA, Washington, DC, USA) were used to estimate the paleo-coastlines (−130 m than at present) and the paleo-climate surfaces of the exposed seafloor area during the LGM.

## 3. Results

### 3.1. Chloroplast DNA Diversity and Population Structure of P. subaequalis

The concatenated cpDNA sequences (*psbC*-*psbZ*, *accD*-*psaI*, and *ndhD*-*psaC*) surveyed across 339 individuals of *P. subaequalis* were aligned with a total length of 1501 bp, including ten substitutions and one indel (6 bp) (Appendix A). Thirteen haplotypes (H1–H13) were identified in the 21 populations of *P. subaequalis* based on these polymorphisms (Figure 1a; Appendix A). Four haplotypes (H2, H3, H4, and H6) are shared by at least two populations, and haplotype H2 is the most common haplotype (found in 12 populations with a frequency of 0.472), while the remaining nine haplotypes are population-specific (Figure 1). At the species level, the cpDNA data revealed high haplotype diversity (*h*_T_ = 0.738) and nucleotide diversity (*π*_T_ = 1.21 × 10^−3^) (Appendix A). At the population level, average intra-population diversity haplotype (*h*_S_) and nucleotide diversity (*π*_S_) were estimated to be 0.088 and 0.136 × 10^−3^, respectively. Population HNZ (Anhui Province) had the highest haplotype diversity (*h* = 0.604), followed by the populations YSH (*h* = 0.441), SYC (*h* = 0.370), LWS (*h* = 0.309), and TJZ (*h* = 0.118) (Appendix A). Population SYC (Zhejiang Province) had the highest nucleotide diversity (*π* = 0.880 × 10^−3^), followed by LWS (*π* = 0.820 × 10^−3^), HNZ (*π* = 0.710 × 10^−3^), YSH (*π* = 0.290 × 10^−3^), and TJZ (*π* = 0.160 × 10^−3^) (Appendix A). The results of linear regression and Pearson’s correlation analysis showed that the population genetic diversity (*h* and *π*) of *P. subaequalis* was not significantly correlated with geographic latitude/longitude at the chloroplast level (*p* = 0.4213–0.9217; Appendix A).

High inter-population genetic differentiation (*G*_ST_ = 0.887) of *P. subaequalis* was detected but had little correlation with the geographic regions (*N*_ST_ = 0.892; *p* = 0.0559 > 0.05), indicating that *P. subaequalis* had strong genetic differentiation even when geographic correlation was weak. AMOVA based on cpDNA data revealed that 87.62% of the species’ total variation in cpDNA was distributed among different populations, with 12.38% explained by variation within populations (Appendix A). A Mantel test conducted on the genetic and geographical matrices found no significant correlation (*r* = 0.069, *p* = 0.213), which indicated that the isolation-by-distance (IBD) effect among *P. subaequalis* populations was minimal.

The cpDNA-rooted network displayed two lineages in the populations of *P. subaequalis* (Figure 1b). In the first lineage (Lineage A), haplotype H2 was located in the central place and was the most widespread haplotype, which thus can be identified as an ancestral haplotype. Haplotype H2 connected with H4, H6, H8, H10, H12, and H13 in a ‘star-like’ shape, each of which differed from H2 by a single substitution. Haplotype H4 further diverged into H3 and H5, and H10 linked to the second lineage (Lineage B), where H7 and H9 were both derived from H11 through a one-step mutation, and H1 was produced by a single mutation in H9. Moreover, haplotype H11, unique to population TXC in Zhejiang Province, was connected to the outgroup (*P. persica*). The cpDNA tree topologies obtained from ML and BI methods yielded a similar topology (Figure 2): all 13 haplotypes were inferred to be monophyletic and were grouped into two subclades (Clade 1 and Clade 2), which was consistent with Lineage A and Lineage B. The divergence time of the two cpDNA lineages (Lineage A and Lineage B) was dated to the late Miocene of the Tertiary period (mean: 11.47 Mya; 95% HPD: 4.59–19.03 Mya), whereas the diversification of various haplotypes occurred in the Quaternary period (ca. 2.58 Mya) (Figure 2).

### 3.2. EST-SSR Diversity and Population Structure of P. subaequalis

All 16 EST-SSR loci were employed for the subsequent analysis given that they were reliable, based on the fact that they did not display significant deviations from the Hardy–Weinberg Equilibrium (HWE) and Linkage Disequilibrium (LD). A total of 74 alleles (*N*A) were identified over the 16 EST-SSR loci among the 410 individuals from 21 populations, with *n* ≥ 6 surveyed and two to eight alleles per locus (Appendix A). The observed heterozygosity (*H*_O_) and expected heterozygosity (*H*_E_) of these loci ranged from 0.041 to 0.624 and 0.059 to 0.582, with mean values of 0.407 and 0.428, respectively; the total genetic diversity (*H*_T_) varied from 0.062 to 0.581, with an average of 0.428; the polymorphism information content (PIC) of the loci ranged from 0.058 to 0.506, with a mean value of 0.371; the genetic differentiation coefficient (*F*_ST_) ranged from 0.100 to 0.325 and averaged 0.169; and the inbreeding coefficient (*F*_IS_) ranged from −0.386 to −0.003, with an average of −0.119 (Appendix A). At the population level of *P. subaequalis*, average estimates of the *N*A, *H*_O_, *H*_E_, *H*_T_, allelic richness (*R*_S_), and inbreeding coefficient (*F*_IS_) were 40, 0.406, 0.360, 0.369, 2.146, and −0.103, respectively (Appendix A). A higher level of genetic diversity was observed in the populations of the Dabie Mountain region in Anhui Province (also the geographic center of *P. subaequalis*), among which the population HNZ harbored the highest *N*A and *R*_S_.

Structure yielded the highest likelihood when individuals were clustered into three groups (*K* = 3, Figure 3a), but the distribution of different genetic components did not recover separate groups that were divided by the geographic barrier of the region, which indicated the existence of genetic mixing (Figure 3b). A similar result was generated by the corresponding PCoA: populations SYC, SLG, LWS, QSW, QL, WFS, DXG, HNZ, and SJD clustered in Group A (red gene pool); TXC, DLX, GDS, SJW, TX, and WLS constituted Group B (green gene pool); and the remaining populations ZXC, LHJ, YSH, TTS, JCY, and TJZ formed Group C (blue gene pool) (Appendix A). Mantel tests of IBD revealed no significant correlation between geographical and genetic distances (*r* = 0.065, *p* = 0.282). According to AMOVA, 93.69% of the genetic variation was within populations, and merely 3.98% was contributed by differences among the 21 populations (Appendix A).

### 3.3. Demographic History of P. subaequalis

Population demographic histories inferred by cpDNA data revealed that mismatch distributions of all populations were multimodal (Appendix A), indicating that *P. subaequalis* experienced a demographic contraction/bottleneck event. Tajima’s *D* was non-significant positive (0.13063, *p* = 0.564), supporting the theory that *P. subaequalis* experienced demographic contraction; however, the Fu’s *FS* was non-significant negative (−1.48232, *p* = 0.357) in the entire population of *P. subaequalis*, suggesting *P. subaequalis* might experience a demographic expansion. However, there were inconsistencies between the results of Fu’s *FS*, Tajima’s *D*, and mismatch distribution analysis in our study, which may be due to the insufficient information contained in the cpDNA fragments.

The bottleneck analyses were based on the models IAM, SMM, and TPM employing EST-SSR loci and the populations TXC, SLG, QSW, ZXC, LHJ, QL, TTS, and JCY; under the Mode-Shift model, the populations TXC, ZXC, and LHJ had normal L-shaped distributions (Appendix A); the above results showed that *P. subaequalis* had experienced a severe bottleneck.

### 3.4. Present, Past, and Future Ecological Niches of P. subaequalis

The ENMs for *P. subaequalis* showed high predictive modal performance (AUC values = 0.989 ± 0.010). The jackknife test on the contribution of bioclimatic variables to the distribution of *P. subaequalis* indicated that temperature seasonality (BIO4), max temperature of warmest month (BIO5), precipitation of driest quarter (BIO17), and precipitation of warmest quarter (BIO18) were the dominant climatic factors influencing and limiting the distribution of *P. subaequalis*. The response curves of the above four bioclimatic variables that limited the distribution of *P. subaequalis* were presented in Appendix A.

The present potential distribution of *P. subaequalis* was largely consistent with its actual scattered and fragmental distribution, apart from some predicted regions where it does not occur at present (e.g., some coastal areas in eastern China) (Figure 4a). Paleo-distribution modeling showed that the suitable habitats for *P. subaequalis* expanded during the periods of MH and LGM but contracted during the LIG. Specifically, during the LGM, the area of potentially suitable habitat of *P. subaequalis* reached a maximum compared to the other periods, with an overall continuous distribution trend expanding eastwards and northwards (Figure 4). In the context of future global climate change, the ENM simulated different reductions in the area of suitable habitat of *P. subaequalis* under the four future greenhouse gas emission scenarios (RCP2.6, RCP4.5, RCP6.0, and RCP8.5) (Figure 5). Furthermore, the core distribution ranges of *P. subaequalis* during the above periods were all predicted in the Dabie Mountain and Yellow Mountain in Anhui Province and the Tianmu Mountain and Simin Mountain in Zhejiang Province.

## 4. Discussion

### 4.1. Genetic Diversity and Differentiation of P. subaequalis Populations

Our cpDNA data revealed high haplotype diversity (*h*_T_ = 0.738) across the 21 populations of *P. subaequalis* in East Asia, which is higher than the average level of chloroplast genetic variation for the 175 plants (*h*_T_ = 0.670) reviewed by Petit et al. (2005) [28], but slightly lower than that of other widely distributed relict tree species, e.g., *Cercidiphyllum japonicum* (*h*_T_ = 0.757) [65]; *Quercus glauca* (*h*_T_ = 0.767) [6]; *Euptelea pleiosperma* (*h*_T_ = 0.893) [66]; and *Liquidambar formosana* (*h*_T_ = 0.909) [67]. The long evolutionary history of *P. subaequalis* may have resulted in the accumulation of such high genetic diversity. Moreover, the highly heterogeneous and fragmented habitat of this species also contributes to the increased frequency of events such as segregation or genetic drift in *P. subaequalis* populations, which in turn maintains the high level of genetic diversity in *P. subaequalis*. At the population level, chloroplast genetic diversity varied considerably among the populations of *P. subaequalis* (*h*, 0.000–0.604; *π* × 10^−3^; 0.000–0.880; Appendix A): populations in the Dabie Mountain and Yellow Mountain in Anhui Province and the Tianmu Mountain and Simin Mountain in Zhejiang Province (HNZ, YSH, TJZ, LWS, and SYC) have higher haplotype diversity. The population genetic differentiation index (*G*_ST_ = 0.887) also indicated large population differences. Considering cpDNA is maternally inherited in *P. subaequalis*, its gene flow is mainly mediated by the seeds of the row ejection mechanism produced by its capsule [17,68]. However, the limited ejection capacity of its seed leads to a smaller dispersal area and limited spreading capacity. Moreover, geographical vicariance and habitat heterogeneity and fragmentation caused by mountains, rivers, and ravines further limit seed flow and thus maintain and deepen the high genetic differentiation in *P. subaequalis* at the cpDNA level. Similar isolation mechanisms are also found in other relict woody plants (e.g., *Davidia involucrata*, *Euptelea pleiosperma*, *Liquidambar formosana*, *Emmenopterys henryi*, and *Tetracentron sinense*) and the companion plant groups in the same climatic zone as *P. subaequalis* (mainly in the subtropical regions of China, e.g., *Quercus glauca* and *Quercus variabilis*) [66,67,68,69,70]. Meanwhile, a lower genetic diversity of *P. subaequalis* was revealed (*H*_E_ = 0.360) by the more evolutionarily conservative characteristic—EST-SSR markers [71]—which was significantly lower than that of perennials (*H*_E_ = 0.680) and cross-pollinated plants (*H*_E_ = 0.650) reported by Nybom (2004) [72] and also lower than that of other reported relict woody plants, e.g., *Disanthus cercidifolius* Maxim. var. *longipes* Chang (*H*_E_ = 0.433) [73]; *Liquidambar formosana* (*H*_E_ = 0.621) [74]; *Liriodendron tulipifera* (*H*_E_ = 0.653) [75]; *Liriodendron chinense* (*H*_E_ = 0.678) [75]; and *Emmenopterys henryi* (*H*_E_ = 0.671) [76].

In addition, a more significant population genetic differentiation was detected based on cpDNA variation compared to EST-SSRs at the range-wide scale (*G*_ST_ = 0.887 vs. *F*ST = 0.0398), which may be explained by a less limited seed flow compared to pollen flow for *P. subaequalis*. Similar research results and mechanisms have also been observed in the woody plants of *Firmiana danxiaensis* [77], *Liquidambar formosana* [67], and *Quercus acutissima* [78].

### 4.2. Population Demography and Multiple Glacial Refugia of P. subaequalis

The population demographic history of *P. subaequalis* inferred by cpDNA data indicated that *P. subaequalis* experienced a recent demographic contraction/bottleneck event. The bottleneck analyses based on EST-SSR loci also indicated that *P. subaequalis* had experienced a recent severe bottleneck. The divergence time of two cpDNA lineages of *P. subaequalis* was dated to the late Miocene of the Tertiary (mean: 11.47 Mya; 95% HPD: 4.59–19.03 Mya) when the uplift of the Himalayan–Tibetan Plateau, the aridification of Central Asia, and the formation of Asian monsoons in the middle and late Tertiary occurred [79,80]. Similar genealogical divergence time was reported in other relict tree species, such as *Tetracentron sinense* (9.60 Mya) [81], *Quercus glauca* (9.07 Mya) [6], and *Liquidambar formosana* (10.30 Mya) [67]. The diversification of different haplotypes of *P. subaequalis* occurred in the Quaternary (ca. 2.58 Mya) (Figure 3), which coincides with the timing of Quaternary climate oscillations.

The Quaternary glaciation was the most recent major geological event in Earth’s history. During the alternation of glacial and interglacial periods, glacial refuges provide suitable ecological niches for many climatic and geographical relict species, and populations in refuges often have high genetic diversity, a rich variety of endemic haplotypes, and ancient ancestral haplotypes [82,83,84,85]. Evidenced by our molecular data and ecological niche modeling, the Dabie Mountain and Yellow Mountain in Anhui Province and the Tianmu Mountain and Simin Mountain in Zhejiang Province were inferred to be multiple glacial refugia of *P. subaequalis* in EAF, which supported the existence of multiple tree refugia (MR hypothesis) in the *Metasequoia* flora of subtropical China. One of the glacial refuges for many other relict tree species is also located in the above mountains, such as *Ginkgo biloba* (Tianmu Mountain) [86], *Cercidiphyllum japonicum* (Tianmu Mountain) [65], *Quercus glauca* (Dabie Mountain and Tianmu Mountain) [6], and *Liquidambar formosana* (Yellow Mountain) [67]. These mountains are also hotspots of biodiversity in East Asia; the complex and varied topography of these mountains and the favorable hydrothermal conditions of the local habitats block and buffer the effects of climatic oscillations of the glacial and interglacial periods, thus well preserving these endangered and relict tree species [87,88].

### 4.3. Response to Climatic Oscillations and Conservation Implications for P. subaequalis

Accompanied by climatic changes and oscillations, the Eurasian relics experienced repeated range expansions and contractions, after which they were capable of tracking suitable ecological niches in response to their changing habitats, thus undergoing adaptive evolution in the context of global climate change [7,12,13]. Based on the results of ENM, the dominant climatic factors affecting and limiting the distribution of *P. subaequalis* include BIO4, BIO5, BIO17, and BIO18, which indicates that alternating wet and dry seasons, temperature seasonality, and extreme heat intrigued by monsoon climate had a limiting effect on the distribution of *P. subaequalis*. Paleo-distribution modeling suggested that the suitable habitats for *P. subaequalis* expanded during the periods of MH and LGM but contracted during the LIG. Specifically, during the LGM, the area of potentially suitable habitat for *P. subaequalis* reached a maximum compared to the other periods, with an overall continuous distribution trend expanding eastwards and northwards (Figure 4). We assumed that a cold-adapted species, such as *P. subaequalis*, might be forced to migrate to higher altitudes in the mountains during the warmer LIG period; this resulted in the fragmentation and contraction of its suitable habitats. In the warm-temperate/subtropical mountains of China, a similar pattern of ‘glacial expansion-interglacial compression’ has also been reported for *Emmenopterys henryi* [89], *Euptelea pleiosperma* [66], and *Liquidambar formosana* [67]. In the context of future global climate change, the ENM simulated contractions in the area of suitable habitats for *P. subaequalis,* and the south of the Yangtze River will no longer be suitable for the survival of this Tertiary relict ‘living fossil’ tree (Figure 5). Tang et al. (2018) [7] similarly concluded that future climate warming will lead to a reduction in the area of potentially suitable habitat for many Tertiary relict plant taxa.

Taken together, combined with the research results of our molecular data and ecological niche modeling, we propose that it is urgent and crucial to protect the populations in multiple glacial refugia located in the Dabie Mountain and Yellow Mountain in Anhui Province and the Tianmu Mountain and Simin Mountain in Zhejiang Province of East China as ‘Management Units’ [90] for in situ conservation in the future. Moreover, ex situ conservation methods, like collecting the seeds and seedlings from different gene clusters and transplanting them together to establish the seed zone, are also essential and effective for the long-term preservation of the genetic resources of *P. subaequalis*.

### 4.4. Conclusion and Prospect

In this study, we employed both plastid (cpDNA) and nuclear data (EST-SSRs) to illuminate the genetic diversity, structure, and phylogeographic pattern of *P. subaequalis*. In the long evolutionary history, *P. subaequalis* has accumulated high haplotype diversity. Weak gene flow by seeds, geographical isolation, and heterogeneous habitats have led to a relatively high level of genetic differentiation in this species. Paleo-distribution modeling suggested that *P. subaequalis* followed the pattern of ‘glacial expansion-interglacial compression’. All molecular and ENM evidence agreed with the MR hypothesis for the ecological success of *P. subaequalis*. Specifically, the Dabie Mountain and Yellow Mountain in Anhui Province and the Tianmu Mountain and Simin Mountain in Zhejiang Province were inferred to be multiple glacial refugia of *P. subaequalis* in East Asia and were proposed to be protected as ‘Management Units’ in the future conservation strategies.

Overall, our study adds to the understanding of how Tertiary relict trees from subtropical China genetically diverged since the late Tertiary and survived during the subsequent global climate oscillation. However, the less effective information contained in the cpDNA and EST-SSRs limited the inference of the population demography history of *P. subaequalis*. With the advent and rapid development of sequencing technologies and population genomics, we can thus deeply explore the contemporary and historical ecological (climatic, geographical) factors shaping the population genetic diversity, structure, and divergence of Tertiary relict trees.

## Figures and Tables

**Figure 1 plants-14-01754-f001:**
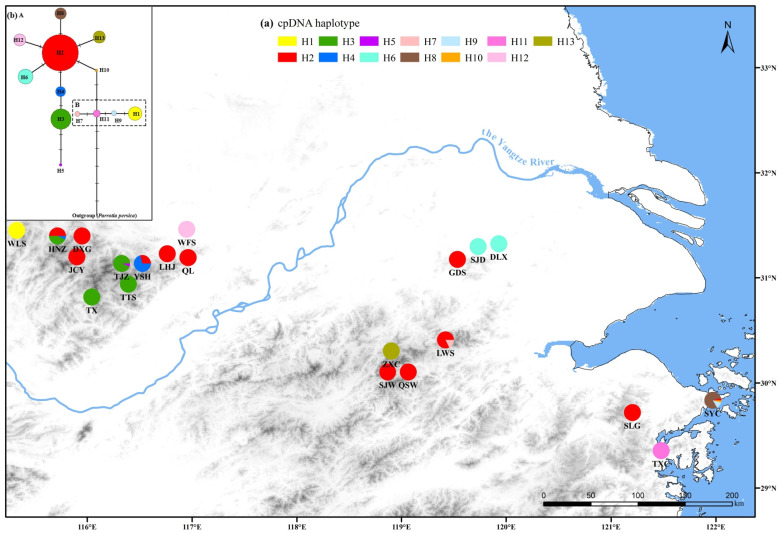
**Distribution of chlorotypes in 21 populations of *Parrotia subaequalis***. (**a**) The 13 chlorotypes in 21 populations of *Parrotia subaequalis*; (**b**) Network of genealogical relationships between the 13 chlorotypes. Circle size represents the frequency of the chlorotypes.

**Figure 2 plants-14-01754-f002:**
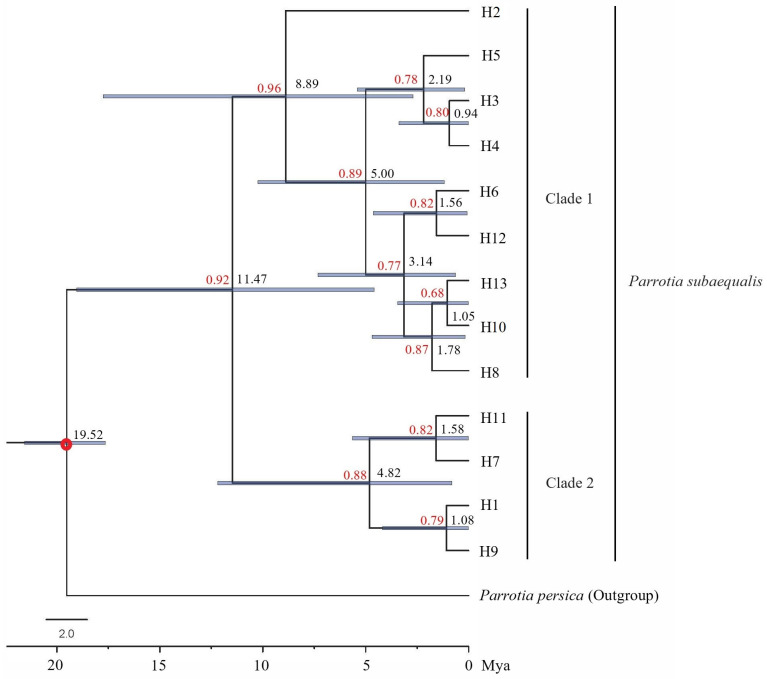
Phylogenetic tree and divergence time of 13 chlorotypes of *Parrotia subaequalis*, based on the combined analysis of three cpDNA sequence (*psbC*-*psbZ*, *accD*-*psaI*, and *ndhD*-*psaC*) data using BI method. The red dot represents the fossil calibration, the red number represents the posterior probabilities, the black number represents the mean value of the divergence time, and the blue bar on each node indicates 95% highest posterior density (HPD) confidence intervals for divergence time estimates.

**Figure 3 plants-14-01754-f003:**
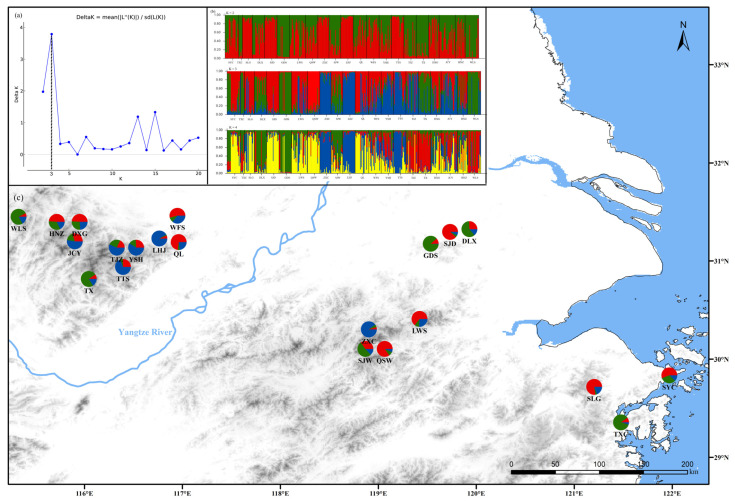
**The results of Structure of *Parrotia subaequalis*.** (**a**) The best value of K for 21 populations of *Parrotia subaequalis*; (**b**) The results of Structure of *Parrotia subaequalis*; (**c**) The geographical distribution of the gene cluster with K = 3.

**Figure 4 plants-14-01754-f004:**
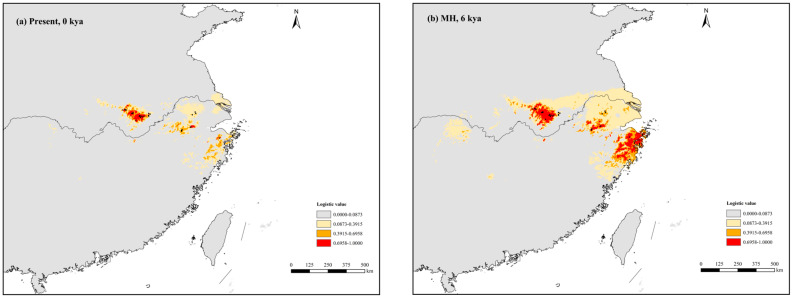
Potential distribution probability (in logistic value) of occurrences for *Parrotia subaequalis* in East China during the different periods. (**a**) At the present (1950–2000); (**b**) in the Mid-Holocene (MH, c. 6 kya); (**c**) at the Last Glacial Maximum (LGM, c. 21 kya); and (**d**) during the Last Interglacial (LIG, c. 130 kya). Presence records of *Parrotia subaequalis* in China (*n* = 31) are plotted as black dots in the maps.

**Figure 5 plants-14-01754-f005:**
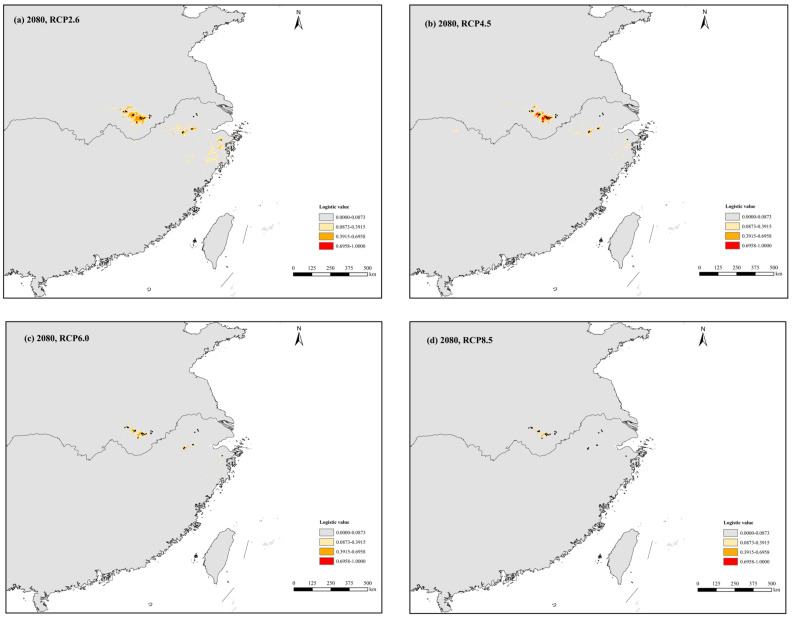
Potential distribution probability (in logistic value) of occurrences for Parrotia subaequalis in East China in the future under the 4 scenarios of carbon dioxide concentration emission. (**a**) RCP2.6; (**b**) RCP4.5; (**c**) RCP6.0; and (**d**) RCP8.5. Presence records of *Parrotia subaequalis* in China (*n* = 31) are plotted as black dots in the maps.

## Data Availability

The original contributions presented in this study are included in the article/Appendix A. Further inquiries can be directed to the corresponding author(s).

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
