# Peer review of "Phylogeography and Population Demography of Parrotia subaequalis, a Hamamelidaceous Tertiary Relict ‘Living Fossil’ Tree Endemic to East Asia Refugia: Implications from Molecular Data and Ecological Niche Modeling"

_plants, 2025, doi:10.3390/plants14121754_

Round 1
Reviewer 1 Report
Comments and Suggestions for Authors
The text you've provided discusses the genetic diversity of the species P. subaequalis based on chloroplast DNA (cpDNA) sequencing. It outlines various important statistical measures, such as haplotype diversity and nucleotide diversity, and indicates the presence of multiple haplotypes across different populations.
315 - Genetic Differentiation: The text states that there is high inter-population genetic differentiation (GST = 0.887), but it suggests that this differentiation does not correlate strongly with geographic regions (NST = 0.892; P = 0.0559). The phrasing "indicating that P. subaequalis had no strong phylogeographic structure" might be clearer if it explicitly states that high GST values can sometimes indicate strong genetic differentiation, even when geographic correlation is weak.
346 - The opening sentence states that all 16 EST-SSR loci were "employed for the subsequent analysis," but it may benefit from explicitly stating the reason for their selection, such as their reliability based on not displaying significant deviations from Hardy-Weinberg Equilibrium (HWE) and Linkage Disequilibrium (LD).
Reviewer 2 Report
Comments and Suggestions for Authors
Dear Authors,
I enjoyed reading your manuscript on a relevant ecological and conservation topic. Please find my detailed comments in the attached file.
Best regards

Reviewer 3 Report
Comments and Suggestions for Authors
Zhang et al. Report on the genetic diversity, populations structure and history as well as future distribution of Parrotia subaequalis an endangered tree species. The methods and data analysis are very well apt to cover the topic and the paper is pleasantly written, the results are a very important addition to our understanding of the influence the glacial dynamics on plant populations. I have only some minor comments: 1. Abstract L. 10: should be “weak gene flow by seed” instead of “weak seed flow”. 2. Line 67: write out full species name at first mention in the main text. 3. Line 451: “by the more evolution-451 arily conservative characteristic of EST-SSR markers” provide a reference for this statement (which is completely true of course).
Reviewer 4 Report
Comments and Suggestions for Authors
Report on Phylogeography and Population Demography of a Hamamelidaceous Tertiary Relict ‘Living Fossil’ Tree Endemic to East Asia Refugia (Parrotia subaequalis): Implications from Molecular Data and Ecological Niche Modeling
In this study, the authors performed population genetic analysis of the Living Fossil Tree, Parrotia subaequalis, and Endemic to East Asia Refugia using EST-SSR and cpDDNA, and distribution estimation using an ecological niche model.
The sampling scheme and analytical approach are well established. The efforts made to analyse the number of populations and samples for population genetic analysis are reasonable.
The scheme is well established, and I thought the presentation and interpretation of the results and discussions were reasonable.
Some comments.
L95: The sentences in cpDNA are in capital letters. Please spell them out or try to make it apporppriate form in Journal.
L96-97: The interpretation based on the nuclear DNA analysis is not incorrect, but strictly speaking, we are looking at the effects via both pollen flow and seed dispersal.
L392-397: Regarding the explanation of factors that affect distribution in the ecological niche model, response curves for each climatic variable would be obtained from the MaxENT output. I think it would be more useful for readers if you presented the response to environmental variables that limit the distribution of this rare tree species, although this could be done in an appendix.
Also, please unify the citation style of the references in the main text and reference list to the journal style.
